# Double charge flips of polyamide membrane by ionic liquid-decoupled bulk and interfacial diffusion for on-demand nanofiltration

Bian-Bian Guo [1,3], Chang Liu [1,2,3], Cheng-Ye Zhu[1,2], Jia-Hui Xin[1], Chao Zhang [1,2] ✉, Hao-Cheng Yang [1,2] ✉ & Zhi-Kang Xu [1,2] ✉

Fine design of surface charge properties of polyamide membranes is crucial for selective ionic and molecular sieving. Traditional membranes face limitations due to their inherent negative charge and limited charge modification range. Herein, we report a facile ionic liquid-decoupled bulk/interfacial diffusion strategy to elaborate the double charge flips of polyamide membranes, enabling on-demand transformation from inherently negative to highly positive and near-neutral charges. The key to these flips lies in the meticulous utilization of ionic liquid that decouples intertwined bulk/interfacial diffusion, enhancing interfacial while inhibiting bulk diffusion. These charge-tunable polyamide membranes can be customized for impressive separation performance, for example, profound $Cl^-/SO_4^{2-}$ selectivity above 470 in sulfate recovery, ultrahigh $Li^+/Mg^{2+}$ selectivity up to 68 in lithium extraction, and effective divalent ion removal in pharmaceutical purification, surpassing many reported polyamide nanofiltration membranes. This advancement adds a new dimension to in the design of advanced polymer membranes via interfacial polymerization.

Polyamide membranes have been recognized as a class of benchmark separation platform for water[1–3], energy[4], chemical[5,6] and pharmaceutical[7] fields involving ionic and molecular sieving, owing to their ability in harnessing surface charges for selective transport and separation of oppositely charged solutes even with a similar size[8,9]. However, despite extensive achievement in developing charged polyamide membranes, they are normally negatively charged once using commercial piperazine (PIP) and trimesoyl chloride (TMC) as monomers[10,11]. This mainly stems from the hydrolysis of excess acyl chloride groups caused by the insufficient monomer diffusion[12,13]. Although a series of advanced strategies in manipulating monomer diffusion have been exploited[14–19], they suffer from synchronous change of bulk and interfacial diffusion (i.e. either enhanced or suppressed simultaneously), eventually leading to narrow charge tailoring window of polyamide membranes. Therefore, up to date, there is almost no available strategy that decouples bulk/interfacial diffusion during interfacial polymerization for developing polyamide membranes with on-demand surface charges.

Ionic liquids (ILs), one of the most attractive green solvents characterized by non-volatile and viscous attributes, have been widely implemented in a wide range of fields from energy to environmental and catalytic science[20–22]. Beyond the utility of solvents, ILs have also

[1]Key Lab of Adsorption and Separation Materials & Technologies of Zhejiang Province, MOE Engineering Research Center of Membrane and Water Treatment, Department of Polymer Science and Engineering, Zhejiang University, Hangzhou 310058, China. [2]The "Belt and Road" Sino-Portugal Joint Lab on Advanced Materials, International Research Center for X Polymers, Zhejiang University, Hangzhou 310058, China. [3]These authors contributed equally: Bian-Bian Guo, Chang Liu. ✉e-mail: zhangchao7@zju.edu.cn; yanghch@zju.edu.cn; xuzk@zju.edu.cn

been exploited as a new class of phase transfer catalysts for accelerating the reaction efficiency of heterogeneous system[23–25]. The impressive characteristic of IL-based phase transfer catalysts lies in rapidly transporting reactants across the interface from one phase to another, which is in analogs with surfactants. This cross-interface feature may be harnessed to boost interfacial diffusion of monomers during interfacial polymerization. Meanwhile, ILs can leverage their inherently viscous attribute to suppress the bulk diffusion and transport of monomers[26,27]. Consequently, we hypothesize that ILs may be hopeful to decouple synchronous change of bulk/interfacial diffusion and resolve the issue of limited charge regulation of polyamide membranes.

In this work, we discover a facile IL-decoupled bulk/interfacial diffusion strategy that enables double charge flips of polyamide membranes by conducting interfacial polymerization at an interface of alkane and IL/water mixture. Distinct from the conventional alkane-water interface that only produces negatively charged polyamide membranes, our design renders the fabrication of polyamide membranes with all-spectrum charges (named PA-IL membrane) from negative to positive to near-neutral charge (Fig. 1 and Supplementary Fig. 1). Remarkably, the positive surface charges of PA-IL membranes reach zeta potential of +34 mV, as opposed to the conventional polyamide membranes with zeta potential of −25 mV, demonstrating the significant advancement for positive PIP-based polyamide membrane. We also demonstrate that IL-enabled negatively and positively charged PA-IL membranes exhibit ultrahigh $Cl^-/SO_4^{2-}$ selectivity above 470 in sulfate recovery and $Li^+/Mg^{2+}$ selectivity up to 68 in lithium extraction, respectively, both of which are much better than that of conventional counterparts.

## Results

### Design of double charge flips of PA-IL membranes

The double charge flips of polyamide membranes can be achieved by the meticulous choice of IL as co-solvent to regulate the interfacial polymerization, in which conventional alkane-water interface is replaced with the interface of alkane and IL/water mixture (Fig. 2a). Here, 1-butyl-3-methylimidazolium tetrafluoroborate ([Bmim][BF$_4$]) was selected as typical IL because of its unique physicochemical merits. First, [Bmim][BF$_4$], a non-volatile polar solvent, is miscible with water at any proportions for forming homogeneous IL/water mixture[28]. This homogeneous polar mixture can generate a clear and stable interface with the nonpolar alkane, which is beneficial for creating defect-free polyamide membranes during interfacial polymerization. Second, [Bmim][BF$_4$] possesses the amphiphilic imidazolium cation ([Bmim]$^+$), which can serve as surfactant-like function to accelerate interfacial diffusion of monomers as well as regulate the associated interfacial polymerization process. Third, [Bmim][BF$_4$] is a viscous molten salt with a higher viscosity than that of water. Such a high viscosity exhibits better suppression in bulk monomer diffusion compared with pure water system. Consequently, by adjusting the dosage of [Bmim][BF$_4$], the bulk and interfacial diffusion during interfacial polymerization can be decoupled for achieving inhibited bulk diffusion and enhanced interfacial diffusion simultaneously, enabling the fabrication of polyamide membranes with on-demand surface charges using PIP and TMC without the need of complicated post-modification methods.

Figure 2b shows obvious and unexplored double charge flips of PA-IL membranes varying with IL proportions, following the sequence of negative-positive-neutral-negative under a neutral pH value. At a low IL content of below 5 v/v%, the PA-IL membrane is negatively charged. With extending IL proportion to 10 v/v%, the surface charge of PA-IL membrane has the first flip from negative to positive. Further increasing IL proportion, the intensity of positive surface charge rises gradually to a peak with zeta potential of +34 mV at 40 v/v% IL and subsequently drops down closing to the neutrally charged state (~+1.4 mV) at 80 v/v% IL proportion. Interestingly, the PA-IL membrane would exhibit the second charge flip and come back to be negatively charged when the IL proportion is 100 v/v%.

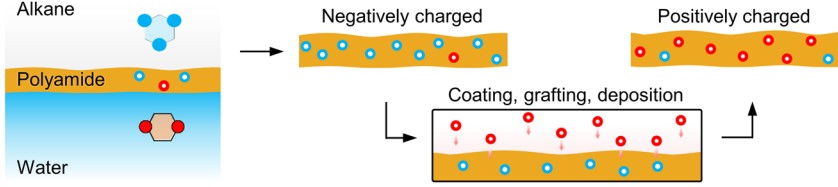

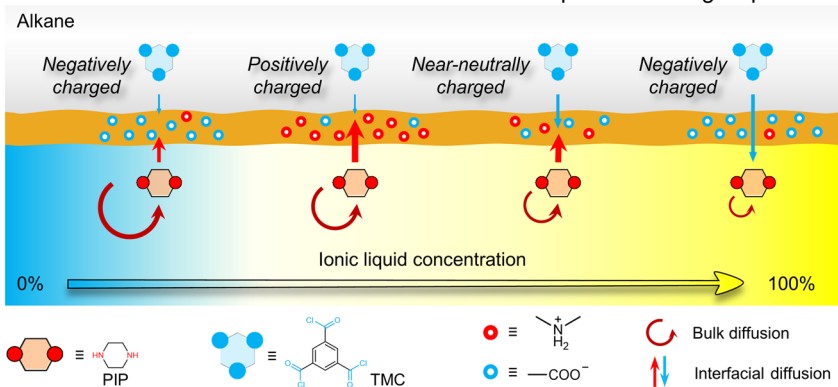

**Fig. 1 | Design concept of double charge flips of polyamide membranes.**
**a** Schematic illustration of negatively charged polyamide membranes resulting from conventional interfacial polymerization (IP) at the alkane-water interface. The surface charge of these membranes can be altered through multi-step post-modifications. **b** Schematic illustration of double charge flips of polyamide membranes by ionic liquid (IL)-decoupled bulk/interfacial diffusion during interfacial polymerization. These polyamide membranes via IL-decoupled strategy present double charge flips: surface charge transits following a negative-positive-near-neutral-negative tendency by controlling IL content.

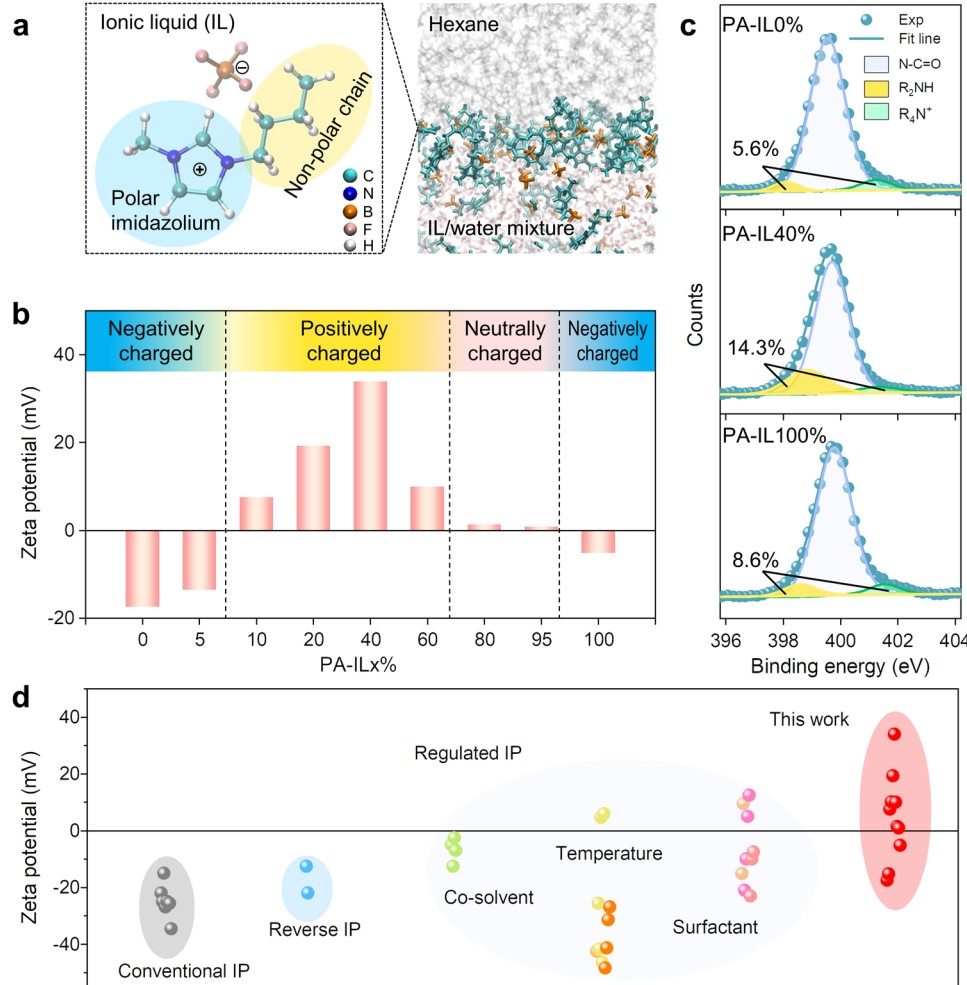

**Fig. 2 | Charge formation and characterization of polyamide membranes via IL-decoupled strategy. a** Chemcial structure of ionic liquid (IL), [Bmim][BF$_4$] and schematic presentation of the interface between IL/water solution and hexane solution. Water and hexane molecules are represented with red and gray colors. The [Bmim]$^+$ cation and [BF$_4$]$^-$ anion of [Bmim][BF$_4$] are represented with cyan and orange colors, and they display apparent accumulation at the interface between IL/water solution and hexane solution. **b** Zeta potential at pH 6 of PA-IL membranes formed by PIP and TMC under different IL volume proportions. The blue, yellow, and pink regions represent negative, positive, and neutral chargeability of PA-IL membranes, respectively. **c** Deconvoluted N 1s spectra on the surfaces of these PA-IL0%, PA-IL40%, and PA-IL100% membranes. The amino group ratios of PA-IL membranes rise and then fall with IL volume proportions. **d** Surface charge windows of polyamide membranes formed by PIP and TMC via different interfacial polymerization (IP) processes. Those reported data from different literature are represented with distinguishing colors and the surface charges of PA-IL membranes are marked in red.

The distinct charge flip phenomenon of PA-IL membranes can be further reflected by investigating the evolution of their surface chemistry (Supplementary Figs. 2 and 3). Figure 2c shows that the amino group content of PA-IL membranes increases sharply from 5.6% to 14.3% with the increase of IL content from 0 v/v% to 40 v/v%, and then decreases to 8.6% as the IL content further increases to 100 v/v%. Distinct from the change tendency of amino group content with the IL proportion, the carboxyl group content on the surface of PA-IL membranes is relatively constant (Supplementary Table 2). And not only that, the content of amino groups in the inner PA-IL40% membrane is higher than that on the surface, while the content of carboxyl groups varies reversely as demonstrated by the elemental depth profile, further verifying the rich amino groups on the positive PA-IL membranes (Supplementary Fig. 4 and Supplementary Table 3). Consequently, the chemical composition of PA-IL membranes experiences an evolution from carboxyl-rich to amino-rich, and then back to carboxyl-rich, which is in good agreement with the tendency of double charge flips.

In addition to the distinct double charge flips phenomenon, our PA-IL membranes also display adjustable zeta potential from −17.5 to +34 mV, significantly broader than other reported polyamide membranes derived from PIP and TMC (Fig. 2d). Normally, the potential value of polyamide membrane at neutral pH is negative to −25 mV through conventional interfacial polymerization or reverse interfacial polymerization[10,29,30]. To extend surface charges of polyamide membranes, some advanced strategies have been proposed to regulate interfacial polymerization, including co-solvent[31], temperature[18,32], or surfactant[17,33,34]. Still, these polyamide membranes are imparted with narrow charge window and relatively weak positive charge. For example, the introduction of low temperature during interfacial polymerization endows polyamide membrane with unilateral positive potential of ~+6 mV, and surfactant addition can further improve positive charge of polyamide membrane to a zeta potential of ~+12.5 mV, both of which are half as low as that of PA-IL40% membrane. These contrast results fully demonstrate that IL-decoupled strategy is superior to broaden the surface charge window of polyamide membranes from negative to positive state.

## Mechanism of double charge flips on PA-IL membranes

To reveal the mechanism of double charge flips on the surface of PA-IL membranes, we investigated the impact of IL on the bulk and

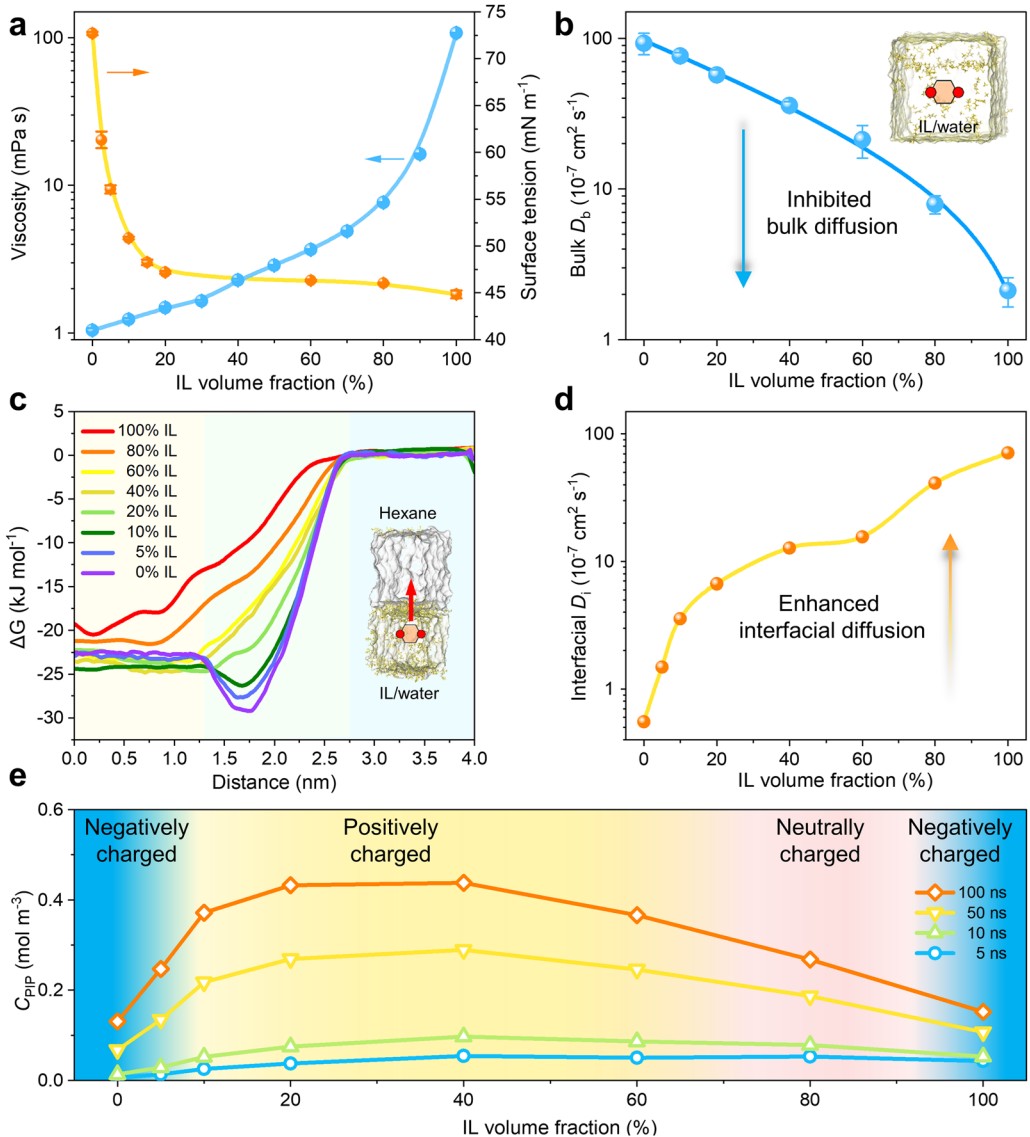

**Fig. 3 | IL-decoupled bulk/interfacial diffusion mechanism. a** Viscosity and surface tension of ionic liquid (IL) /water solutions of varied IL volume proportions. The error bars are standard deviation of three replicate measurements. **b** Theoretical bulk diffusion rates ($D_b$) of PIP molecules in the IL/water solutions of varied IL volume proportions. Insert in **b**: illustration of bulk PIP diffusion in the IL/water solution. The error bars are standard deviation of three replicate calculations. **c**, **d** Interfacial diffusion behavior of PIP molecules. **c** PIP free energy ($\Delta G$) versus the z coordinates perpendicular to the interface of hexane and IL/water solutions. Insert in **c**: schematic illustration of the PIP movement pulled from IL/water solution to hexane solution. The initial coordinate is 2 nm away from the interface in the IL/water side and the total displacement is 4 nm. **d** Simulated interfacial diffusion coefficients of PIP ($D_i$) at varied IL volume proportions. **e** Calculated PIP diffusion amount ($C_{PIP}$) from the bulk IL/water solution to the opposite hexane side at varied IL volume proportions. $C_{PIP}$ is calculated based on the above quantified diffusion coefficients, i.e., $D_b$ and $D_i$. Starting PIP concentration is 100 mol m$^{-3}$ and the cell sizes of hexane and IL/water solutions are both 100 nm ×1000 nm.

interfacial diffusion of monomers during interfacial polymerization. Figure 3a shows that the viscosity of pure IL is around 108 mPa·s, which is 100-fold higher than that of water (1 mPa·s). As the IL was mixed with water, the bulk viscosity of the IL/water solution increases exponentially with the IL content due to high viscosity of IL. This enhanced bulk viscosity is capable of effectively suppressing PIP diffusion behavior during IL/water solution over conventional water system. We further employed molecular dynamics simulation to quantify the PIP diffusion behavior in the IL/water solution. The theoretical bulk diffusion coefficient ($D_b$) of PIP is $3.58 \times 10^{-6}$ cm$^2$ s$^{-1}$ and $2.11 \times 10^{-7}$ cm$^2$ s$^{-1}$ in the 40 v/v% IL and 100 v/v% IL, both of which are much slower than that in pure water with a $D_b$ of $9.30 \times 10^{-6}$ cm$^2$ s$^{-1}$ (Fig. 3b). This difference is expected to imposing significant impact on the growth of polyamide and its surface charge during the interfacial polymerization process[35,36].

In striking contrast to the weaken bulk diffusion, IL is capable of facilitating PIP transport across the interface of alkane and IL/water into the reaction zone. As shown in Fig. 3c, the free energy barriers of PIP diffusion ($\Delta G$) reduce with the increase of IL content across the interface of alkane and IL/water solution. On the basis of energy difference between the maximum and minimum $\Delta G$ values, the interfacial diffusion coefficient ($D_i$) of PIP across the interface of alkane and IL/water mixture can be calculated in Fig. 3d. The $D_i$ values are improved by more than one order of magnitude from $5.52 \times 10^{-8}$ cm$^2$ s$^{-1}$ for 0 v/v% IL to $1.27 \times 10^{-6}$ cm$^2$ s$^{-1}$ for 40 v/v% IL (Supplementary Table 4). Such facilitated interfacial diffusion can be also manifested by more than 10 times higher amount of PIP across the interface of hexane and 40 v/v% IL than that at the hexane-water interface (Supplementary Fig. 6). The IL-facilitated interfacial diffusion can be further understood from the perspective of molecular level. In detail, due to its

amphiphilic feature, the [Bmim]$^+$ of IL accumulates at the interface of alkane and IL/water preferentially and leverages its alkyl substituents for inserting into alkane, as validated by the sharply reduced surface tension of IL/water solution even with a low IL content (Fig. 3a and Supplementary Table 5). When the [Bmim]$^+$ accumulation at the interface is saturated, the IL content entering bulk IL/water solution rises with further adding IL proportion (Supplementary Figs. 7 and 8). In addition to the enhanced interfacial transportation of PIP, IL can also improve TMC diffusion from alkane to IL/water solution. Particularly, IL-rich region favors TMC partitioning with promoted solubility and restricted hydrolysis, as evidenced by a widening of TMC equilibrium distribution around the interface with increased IL content (Supplementary Figs. 9 and 10).

Our in-depth analysis of both bulk and interfacial diffusion reveals that the IL effectively modulates the two diffusion processes of PIP in tandem, instead of merely amplifying one at the expense of the other, as is often observed with typical strategies. We continued to examine the whole journey of PIP diffusion invoking numerical simulations, encompassing both its bulk diffusion and transport across the interface between alkane and the IL/water solution. Initially at 5 ns, the PIP diffusion amounts ($C_{PIP}$) monotonously rise with IL content, which is in consistent with enhanced interfacial diffusion (Fig. 3e). This indicates that interfacial diffusion plays a dominance on the $C_{PIP}$ at the initial stage. With extending time scale, the IL-suppressed bulk PIP diffusion begins to restrict the $C_{PIP}$, therefore resulting in a peak $C_{PIP}$ value of the 40 v/v% IL at 50 ns, which is 4.3 and 2.7 times than that of 0 v/v% IL and 100 v/v% IL (Fig. 3e and Supplementary Fig 11). At this point, the $C_{PIP}$ variations under varying IL contents well match with the surface charges of PA-IL membranes.

The IL-modulated diffusion behavior of PIP inevitably impacts the polymerization rate, which, in turn, affects the structure of polyamide and its subsequent charge properties. To gain deep insight into the interplay between diffusion and reaction, we formulated a simplified kinetic model of interfacial polymerization using a set of differential equations (Eqs. (1)–(3)):

$$\frac{dc_{amide}}{dt} = k_r c_{PIP} c_{TMC} \tag{1}$$

$$\frac{dc_{PIP}}{dt} = \alpha P k_d \left(1 - \frac{c_{amide}}{c_{amide} + c_{TMC}}\right)^\beta (c_{wPIP} - c_{PIP}) - k_r c_{PIP} c_{TMC} \tag{2}$$

$$\frac{dc_{TMC}}{dt} = -k_r c_{PIP} c_{TMC} \tag{3}$$

where $c_{amide}$, $c_{PIP}$, $c_{wPIP}$, and $c_{TMC}$ denote the concentrations of the resulting amide, PIP in the reaction zone, initial PIP in solution, and TMC in the reaction zone, respectively. $\alpha$ represents a scaling factor, reflecting the supplementary rate difference of the bulk diffusion process to the initial interfacial diffusion concentration (Detailed exposition in the Supplementary Information). Our computational insights reveal that the rate coefficient for the reaction between PIP and TMC ($k_r = 6.15 \times 10^8$ s$^{-1}$ M$^{-1}$) is roughly on par with the interfacial diffusion rate coefficient of PIP ($k_d = 0.4 \times 10^8$ s$^{-1}$ ~ $17.3 \times 10^8$ s$^{-1}$), thus facilitating the instantaneous polymerization of monomers in the reaction zone to form the polyamide layer. This is validated by the concentration variations during interfacial polymerization observed in our model (Supplementary Figs. 12 and 13). Specifically, the plateau in these curves suggests an emergence of a self-limiting effect, indicating that the nascent polyamide layer is swiftly formed. With this rapid reaction rate, the relative concentrations of both monomers in the reaction zone are primarily governed by the interfacial diffusion of PIP. Further inspecting these curves in different IL contents, we note that the concentration of PIP initially surges and then recedes. When the

volume fraction of IL ranges between 20 v/v% and 40 v/v%, the concentration of PIP surpasses that of TMC, suggesting that the resultant polyamide layer in this concentration range likely possesses a substantial number of unreacted amino groups, thus presenting an obviously positive charge. This theoretical observation aligns well with the observed trends in membrane charge properties.

Building on this analysis, we proposed a decoupled dual diffusion mechanism that involves enhanced interfacial diffusion and inhibited bulk diffusion of monomers simultaneously to uncover the double charge flips phenomenon. On the one hand, significant enhancement in interfacial diffusion of PIP leads to a drastic excess of PIP monomers in the reaction zone, which tends to create an amino-terminated polyamide surface. As a result, the surface charges on PA-IL membranes gradually turn from inherently negative to positive even at low IL proportions. On the other hand, the restricted bulk diffusion of PIP meanwhile enhanced interfacial TMC diffusion at high IL proportions reduces the PIP excess in the reaction zone. Consequently, the amino groups of PIP are sufficiently reacted with TMC during interfacial polymerization, giving rise to diminished positive charge of PA-IL membranes and even neutral charge at high IL proportions. Further increasing IL proportions, the pronounced TMC diffusion would play a pivotal role in initiating the secondary charge inversion, resulting in a negatively charged polyamide surface due to the abundance of hydrolyzed carboxyl groups.

More intriguingly, our decoupled bulk/interfacial diffusion mechanism to create positive charge of polyamide membrane is generic and can be also extended to other functional molecules. First, we explore this mechanism using 1-ethyl-3-methylimidazolium tetrafluoroborate ([Emim][BF$_4$]), a molecular analog of [Bmim][BF$_4$], as co-solvent to execute the similar interfacial polymerization. Remarkably, at a concentration of 40 v/v%, [Emim][BF$_4$] also induces a charge flip on the polyamide membrane, although less pronounced than with [Bmim][BF$_4$] (Supplementary Fig. 14). This variation is attributable to the physicochemical properties of [Emim][BF$_4$], exhibiting slightly higher surface tension ($57.5 \pm 0.3$ mN m$^{-1}$) and lower viscosity ($1.72 \pm 0.02$ mPa·s). To further extend universality, we substituted IL with a combination of sodium dodecyl sulfate (SDS) and glycerol as additives in the interfacial polymerization, in which the former is capable of facilitating interfacial diffusion via the reduced interfacial tension and the latter can harness its intrinsic viscosity to suppress bulk diffusion. This approach also successfully achieves a charge flip from negative to positive on the polyamide membrane (Supplementary Fig. 14). These results strongly validate the decoupled bulk/interfacial diffusion mechanism as a reliable method for modulating the surface charge of PIP-based polyamide membranes.

## Structures and properties of PA-IL membranes

Beyond fine regulation of surface charge, the IL-decoupled bulk/interfacial diffusion strategy also has a large impact on the surface morphologies and inner structures of polyamide nanofilms on the PA-IL membranes (Fig. 4). The PA-IL nanofilms are gradually flat and smooth accompanied with reduced surface roughness with increased IL content as the viscosity increase of IL/water solutions, stemming from the improved interfacial stability during interfacial polymerization[37] (Supplementary Figs. 15 and 16). Much more interesting, the thickness of PA-IL nanofilms increases sharply from 130 nm to 483 nm with the increase of IL content from 0 v/v% to 10 v/v% and then declines to 132 nm when the IL content further increases to 40 v/v% (Fig. 4a and Supplementary Fig. 17). This IL-regulated thickness of PA-IL nanofilms can be also revealed from the view of a competing dual diffusion mechanism. At a small amount of IL, the facilitated PIP diffusion across the interface would boost the interfacial polymerization rate, resulting in a much thicker nanofilm. However, with a high IL content, the PIP diffusion into the reaction zone is substantially

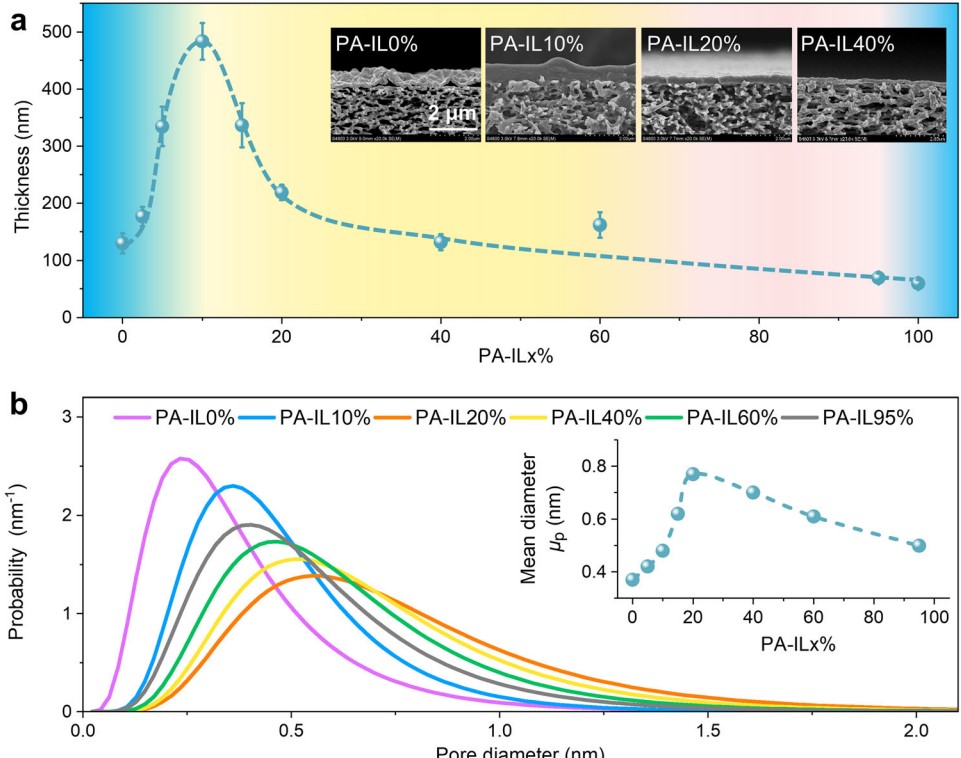

**Fig. 4 | Surface morphologies and inner structures of PA-IL membranes. a** The thickness of polyamide nanofilms on the PA-IL membranes prepared under different ionic liquid (IL) volume proportions. Insert in **a**: cross-sectional SEM images of PA-IL membranes with top polyamide nanofilm composited onto porous PES substrate. The thickness of PA-IL nanofilms increases rapidly first and then gradually declines. The error bars are standard deviation of three replicate measurements. **b** The pore sizes and disrtibution of PA-IL membranes prepared under different IL volume proportions. The pore sizes are analyzed with neutral PEG molecules of different molecular weights as probes. Insert in **b**: The mean pore diameter varaition of PA-IL membranes with the IL content.

restricted, slowing down the interfacial polymerization rate and thus producing a thinner nanofilm.

To reflect the utility of IL-decoupled bulk/interfacial monomer diffusion on the inner structures, we employed different molecular weights of PEG and saccharides as probes for evaluating pore size and molecular weight cutoff. As illustrated in Fig. 4b, the pore sizes of PA-IL membranes increase initially and then decrease with an estimated PEG MWCO ranging between 256 and 669 Da (Supplementary Fig. 18). This range is broader than that observed for the pristine PA-IL0% membrane exhibiting a PEG MWCO of 212 Da. These findings suggest that PA-IL membranes possess a relatively more open network structure with enlarged pore sizes, significantly influenced by the IL-driven bulk/interfacial diffusion competition. Briefly, at a high IL content, although possessing the elevated PIP amount in the reaction zone by the enhanced interfacial diffusion, there is relatively balanced monomer ratios caused by the suppressed bulk diffusion, results in generating complete polyamide network with enlarged pore sizes.

### Performance of on-demand nanofiltration

In combination of IL-enabled well-tuned surface charges and pore structures, PA-IL membranes can be customized for impressive nanofiltration performance towards diverse application scenarios, which are not almost achievable with the conventional polyamide membranes. Figure 5a illustrates that these PA-IL membranes display distinct ion screening performance mainly dictated by the synergistic effect of size sieving and Donnan exclusion. The salt rejection of the PA-IL0% membrane follows the order of $Na_2SO_4$ (98.5%) >$MgSO_4$ (97.7%) >$MgCl_2$ (89.3%) >NaCl (54.1%), exhibiting the typical separation feature of negatively charged polyamide membranes. The relatively

high retention of $Mg^{2+}$ (Stokes radius of 3.47 Å) is ascribed to the size exclusion caused by the small pore diameter (~0.32 nm) of the PA-IL0% membrane. However, the salt rejection order turns to $MgCl_2$ (95.1%) >$MgSO_4$ (66.8%) >NaCl (59.9%) >$Na_2SO_4$ (31.6%) for PA-IL40% membrane, featuring a much higher retention of $MgCl_2$ than that of $Na_2SO_4$. This is because PA-IL40% membrane displays strong and stable electrostatic repulsion for divalent cation $Mg^{2+}$ owing to its positively charged surface that elevates the hindrance of $Mg^{2+}$ into the nanopores. In addition to the positive surface charge, PA-IL40% membrane also possesses rich secondary amino groups charged sites in the inner polyamide layer, which also increases the permeation resistance of $Mg^{2+}$. As a result, PA-IL40% membrane gives a higher $MgCl_2$ rejection compared to PA-IL0% membrane. Distinct from $MgCl_2$ rejection, positively charged PA-IL40% membrane has strong electrical attraction of divalent anion $SO_4^{2-}$. Additionally, its loose network also significantly reduces the rejection of $Na_2SO_4$. The synergy of surface charge and pore size endows PA-IL40% membrane with a low $Na_2SO_4$ rejection, even lower than monovalent NaCl. For $MgSO_4$ with equal cationic and anionic charge, its retention mainly depends on the ion diffusivity in the pore channel and correspondingly declines in the loose PA-IL40% membrane. As further weakening the surface charge of PA-IL membranes with high IL proportions, the near-neutrally charged PA-IL95% membrane has a dramatic decrease in rejection to both divalent cations and anions, indicating that size exclusion becomes dominant over electrostatic repulsion during separation. Taken together, the easy tailoring strategy and encouraging performance make it possible to customize polyamide membranes to satisfy diverse nanofiltration applications.

To further demonstrate the application potential, these double charge flips of PA-IL membranes can be harnessed to execute different

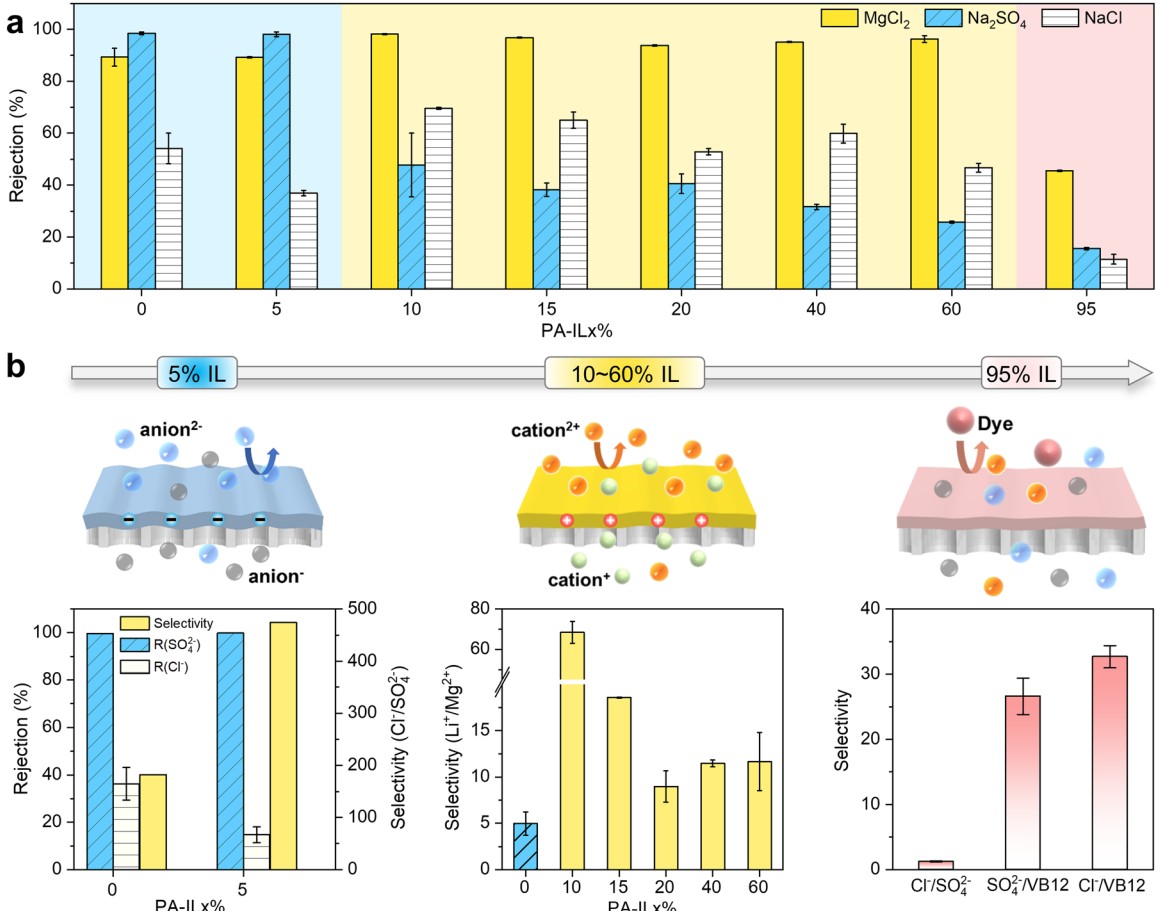

**Fig. 5 | Customized application potential of double charge flips of polyamide membranes. a** Filtrate performance of PA-IL membranes formed by PIP and TMC at varied IL volume proportions. Inorganic salt solutions (Na$_2$SO$_4$, MgCl$_2$, or NaCl) of 1.0 g L$^{-1}$ were applied as the feed at 6 bar. The salt rejection orders depend on the surface charge of PA-IL membranes, therefore, negatively, positively, and near-neutrally charged PA-IL membranes are represented with the blue, yellow, and pink regions, respectively. The error bars are standard deviation of three replicate measurements. **b** Schematic illustrations of on-demand applications of PA-IL membranes and corresponding selective separation performance: Cl$^-$/SO$_4^{2-}$ selectivity of negatively charged PA-IL membranes prepared with ionic liquid (IL) volume proportion less than 5 v/v%. Binary salt solution containing 0.5 g L$^{-1}$ Na$_2$SO$_4$ and 0.5 g L$^{-1}$ NaCl was applied as the feed. Li$^+$/Mg$^{2+}$ separation factors of positively PA-IL membranes were measured at 6 bar using the binary salt solution consisting of 2.0 g L$^{-1}$ MgCl$_2$ and 0.1 g L$^{-1}$ LiCl as feed solution. Ion/pharmaceutical selectivity of neutrally charged PA-IL95% membrane was evaluated with the mixture of 0.1 g L$^{-1}$ model pharmaceutical VB12 and 0.5 g L$^{-1}$ inorganic salt (Na$_2$SO$_4$ or NaCl) as feed solution. The error bars are standard deviation of three replicate measurements.

separation tasks. First, we employed PA-IL5% membrane that features a negatively charged surface and a slightly loose pore structure for removing monovalent anion (Cl$^-$) in sulfate recovery, an important process during the resource utilization of saline wastewater[38]. Figure 5b shows that the PA-IL5% membrane demonstrates improved Cl$^-$/SO$_4^{2-}$ selectivity as high as 470. In contrast, the PA-IL0% membrane has a relatively low Cl$^-$/SO$_4^{2-}$ selectivity of 182, which is below half that of PA-IL5% membrane (Supplementary Table 8). Further comparing with previous reports, the PA-IL5% membrane also delivers Cl$^-$/SO$_4^{2-}$ selectivity exceeding many of the state-of-art negative PIP-based polyamide membranes (Supplementary Fig. 21 and Table 9). Such high Cl$^-$/SO$_4^{2-}$ selectivity of PA-IL5% membrane mainly originates from the facilitated Cl$^-$ permeation and high rejection of SO$_4^{2-}$.

Second, we leveraged the significant positive surface charge of PA-IL membranes for resolving the issue of inferior Li$^+$/Mg$^{2+}$ separation selectivity during lithium extraction in brine lakes. Although nanofiltration-enabled lithium extraction has been widely applied, the conventional polyamide membranes suffer from undesirable Li$^+$/Mg$^{2+}$ screening performance imposed by the negative surface charge[39]. In contrast, our positively charged PA-IL membranes exhibit Li$^+$/Mg$^{2+}$ screening performance over the negative PA-IL0% membrane by the facile control of IL proportions without the need of complicated post-

modifications. The Li$^+$/Mg$^{2+}$ separation factors of PA-IL nanofiltration membranes follow the order PA-IL0% (5) < PA-IL20% (9) < PA-IL40% (12) < PA-IL15% (19) < PA-IL10% (68), regulated by the synergy of electrostatic repulsion and steric hindrance. Notably, the Li$^+$/Mg$^{2+}$ selectivity of PA-IL40% membrane is improved to 2.3 times that of PA-IL0%, which is comparable to that found in common positive PEI-based membrane (Supplementary Table 10). This is because PA-IL40% membrane possesses high positive surface charge that gives rise to strong electrostatic repulsion for resisting Mg$^{2+}$ permeation despite its loose pore structure. For relatively weak positive PA-IL10% and PA-IL15% membranes, they exhibit relatively dense structure with smaller pore size than Mg$^{2+}$, thus allowing for more effective Mg$^{2+}$ rejection. The synergy effect of positive surface and dense network would greatly boost the Li$^+$/Mg$^{2+}$ selectivity. Remarkably, the dense and positive PA-IL10% membrane has up to 13.6 times Li$^+$/Mg$^{2+}$ separation factor than pristine PA-IL0%. The Li$^+$/Mg$^{2+}$ separation factors of these PA-IL membranes exceed commercial nanofiltration membranes and even most literature membranes, despite using conventional monomers (Supplementary Fig. 22). Furthermore, such solid Li$^+$/Mg$^{2+}$ screening performance can be stably sustained for long time as validated by a continuous 300-h cross-flow nanofiltration test (Supplementary Fig. 23).

Third, our PA-IL95% membrane with intact but weakly charged network exhibits poor selectivity of both monovalent and divalent salts (<2) but high retention for various dyes with molecular weight above 450 Da (Supplementary Fig. 24), demonstrating potential application in textile wastewater decoloring. In addition to textile wastewater decoloring, our PA-IL95% membrane can be used in pharmaceutical desalting to remove excess counter ions. We performed the desalination process of organic pharmaceutical solution with VB12 as a model molecule. The result demonstrates that the PA-IL95% efficiently preserves 97.5% VB12 with the $SO_4^{2-}$/VB12 selectivity of 27 and $Cl^-$/VB12 selectivity of 33, indicating that PA-IL95% membrane can not only reduce the pharmaceutical discharge but also meet different desalting requirements.

## Discussion

In conclusion, we present a facile IL-decoupled bulk/interfacial diffusion strategy to achieve the one-step synthesis of customized polyamide membranes with negatively, positively, or near-neutrally charged surfaces at the interface of alkane and IL/water mixture. Under the competing diffusion between enhanced interfacial monomer transport and restricted bulk monomer migration, the diffusion rates of PIP were modulated among a range of orders of magnitude, revealing a significant increase followed by a decrease with the IL content as convinced by molecular simulation and experimental monitoring. The tunable monomer diffusion provides the varied ratios of PIP and TMC in the reaction zone during the IP process for manipulating polyamide membranes with amino-rich or carboxyl-rich compositions, allowing for double charge flips of polyamide membranes meanwhile possessing distinct morphologies and pore structures. The double charge flips of PA-IL membranes can be customized for satisfying different separation demands: negatively charged membrane exhibits high $Cl^-$/$SO_4^{2-}$ selectivity above 470 by slightly enlarging the pore size in sulfate recovery; positively charged membranes efficiently intercept $Mg^{2+}$ with $Li^+$/$Mg^{2+}$ selectivity up to 68 in lithium extraction; the weakly charged membrane is suitable for separation of pharmaceutical and divalent ions under inhibited electrostatic interaction. Moreover, our IL-regulated interfacial polymerization has been proven to be a facile, reliable approach, with great implication of designing and advancing the properties of conventional polyamide membranes based on a deeper insight into underlying diffusion kinetic. Whilst the scope of this study is limited to traditional amine monomers, from a broader perspective, it may inspire the designs of high-performance functional membranes incorporating unusual or stimulus-responsive agents to meet diverse demands in practical applications.

## Methods

### Preparation of polyamide composite membranes

Interfacial polymerization at the interface of alkane and IL/water mixture was conducted on the PES substrate to fabricate the polyamide composite membranes. A certain amount of PIP was dissolved into the aqueous mixture of IL ([Bmim][BF₄]) and water as the polar phase (for example, $vol_{IL}$: $vol_{water}$ = 40:60, defined as 40 v/v% IL solution), and TMC was dissolved at hexane as the non-polar phase. PIP and TMC solutions were sequentially spreading on the PES substrate by a vacuum-assisted process[40]. For example, a PES substrate with a diameter of 5 cm was first placed in the vacuum filtration apparatus. 2 mL PIP solution (8 g L⁻¹, dissolved in 40 v/v% IL solution) was added onto the PES substrate and then was infiltrated within the substrate under vacuum for 30 s. 2 mL TMC solution (2.4 g L⁻¹) was then transferred onto the substrate by a pipette. The IP reaction took place for 60 s, and the residual TMC solution was removed after the reaction. As-prepared polyamide composite membrane was post-treated in a 60 °C oven for 5 min and finally was thoroughly rinsed with DI water three times to remove IL in the PES substrate. It was stored in the 4 °C DI water before further characterization. This polyamide composite membrane was

named PA-IL40% membrane, 40% refers to the IL volume fraction in the PIP solution. Different PA-IL membranes were obtained by varying IL volume fraction from 0 v/v% to 100 v/v%. As for the PA-IL95% membrane, 10 g L⁻¹ PIP and 1.0 g L⁻¹ TMC were especially used to generate an intact polyamide membrane.

### Molecular dynamics simulation

In the simulation of PIP solutions, the employed simulation box was a cube of 50 Å per side, encompassing 20 PIP and solvent molecules, specifically mixtures of IL and water. For biphasic solution systems, a three-tiered cuboid box of dimensions 60 × 60 × 240 Å³ was utilized to fabricate interfaces between hexane and various IL/water mixtures. Arranged from bottom to top, the box contained hexane, IL/water mixture, and hexane respectively, with 50 PIP and 40 TMC molecules incorporated. Atomistic models with full flexibility were grounded in DFT calculations, employing IL parameters based on the OPLS-AA force field derived from Doherty et al.[41–43]. The three-site water model, SPCE, was utilized for water molecules[44]. Three-dimensional periodic boundary conditions (PBC) were used to avoid the influence of the box boundary during simulation. The cut-off distance of non-bonded interactions is 13 Å, while long-range electrostatic interactions were addressed using the particle-mesh Ewald (PME) method[45]. Initial structures for MD simulations were sculpted using the Packmol program[46]. Pre-equalization of the system to dispel excessive initial structural stress was accomplished using the steepest descent algorithm[47]. This was followed by a 10 ns NPT simulation (with a 2-fs time step) to enable convergence to actual density, and then a 50 ns production simulation within the NPT ensemble with a 1-fs time step for data collection. Temperature and pressure were coupled using the v-rescale thermostat and Berendsen barostat at 298 K and 1 atm[48,49]. The range for each hexane-IL/water interface was determined from Z-dependent density according to the widely accepted 90% criterion, whereby the interface position is defined as the coordinate corresponding to 10−90% n-hexane density, calculated from density profiles[50]. Then theoretical bulk diffusion rate constants ($D_b$) of PIP were obtained by linear regression of mean square displacement (MSD) as Eq. (4):

$$D_b = \frac{MSD(t)}{6t} \tag{4}$$

In subsequent free energy calculations, orthogonal simulation boxes of dimensions 40 × 40 × 100 Å³ was constructed, wherein the IL/water mixture occupied the region where z < 50 Å, and hexane occupied the region where z > 50 Å, and a PIP molecule was placed within the water layer with centroid coordinates (20 Å, 20 Å, 20 Å). Initially, energy minimization was conducted, followed by a 5 ns pre-equilibration under the isothermal-isobaric ensemble (NPT) employing the Berendsen barostat[49], during which position restraints were applied to the ions. Steered MD simulations[51] were then performed under the NPT ensemble, with the PIP molecule being pulled by 4 nm within 0.8 ns to traverse the hexane-IL/water interface. From the simulation trajectory, 40 configurations were extracted at equal time intervals, each undergoing 1 ns of NPT pre-equilibration and 10 ns of umbrella sampling simulations. The reaction coordinate was defined as the projection of the distance vector between the PIP's center of mass coordinates from its initial coordinates onto the pulling vector. The free energy profile for the interfacial diffusion of PIP was derived from the potential of mean force along the reaction coordinate using the weighted histogram analysis method (WHAM)[52]. All MD simulations were carried out through GROMACS 2020.6[53]. All visualization structures are provided by VMD 1.9.4[54].

The energy difference between the maximum and minimum values (ΔG) was considered the diffusion energy barrier of PIP transportation across the interface of hexane and IL/water mixture[55]. The

corresponding interfacial diffusion coefficient ($D_i$) and interfacial diffusion rate constant ($k_d$) of PIP from the IL/water mixture to the hexane side was calculated according to Eqs. (5) and (6)[56,57]:

$$D_i = \lambda^2 \frac{k_B T}{h} \exp\left(-\frac{\Delta G}{RT}\right) \tag{5}$$

$$k_d = \frac{k_B T}{h} \exp\left(-\frac{\Delta G}{RT}\right) \tag{6}$$

where $\lambda$ is the interface thickness, $k_B$ is Boltzmann's constant, $h$ is Planck's constant, $R$ is the ideal gas constant, and $T$ is the absolute temperature.

The numerical calculation was performed to evaluate the PIP diffusion amount ($C_{PIP}$) considering above dual diffusion behaviors. In a rectangular model, an IL/water box and a hexane box of 100 nm × 1000 nm were separated by a thin barrier of $\lambda$ thickness. The PIP diffusion coefficients were $D_b$, $D_i$, and $3.5 \times 10^{-5}$ cm$^2$ s$^{-1}$ in the IL/water, interfacial barrier, and hexane, respectively. The initial PIP concentration in IL/water was 100 mol m$^{-3}$, then the PIP distribution was calculated every 0.5 ns for a total of 1.0 μs.

### Determination of pore size distribution
The pore sizes and distributions of PA-IL membranes were analyzed by detecting the rejection of neutral PEG and saccharides with different molecular weights. The effect of hydrodynamic differences on the PEG size is simplified without consideration. The PEG and saccharides concentrations were detected by total organic carbon analyzer (TOC, GE, Sievers 900, USA). The molecular weight cut-off (MWCO) of the PA-IL membrane refers to the molecular weight of the PEG with a rejection of 90%. The mean adequate pore size ($\mu_p$) and its geometric standard deviation ($\sigma_p$) approximate the size of PEG at 50% rejection and the ratio of PEG sizes at the rejection of 50% and 84.13%, respectively[58]. The pore size ($r_p$) and its distribution can be expressed with the probability density function as Eq. (7).

$$\frac{df(r_p)}{dr_p} = \frac{1}{\sqrt{2\pi} r_p \ln\sigma_p} \exp\left[-\frac{\left(\ln r_p - \ln\mu_p\right)^2}{2\left(\ln\sigma_p\right)^2}\right] \tag{7}$$

### Separation performance of polyamide composite membranes
Nanofiltration tests were conducted on a cross-flow filtration system (Hangzhou Saifei, SF-SA, China). The effective separation area is 7.07 cm$^2$ and the applied hydraulic pressure is 6 bar. PA-IL membranes were firstly compacted at 6 bar for 1 h before data collection to achieve stable performance. 1.0 g L$^{-1}$ Na$_2$SO$_4$, MgCl$_2$, or NaCl solution were used as the feed sequentially to evaluate the nanofiltration performance for different salt ions. The feed and permeate were collected at a certain interval at 6 bar. The water flux ($J$) was calculated as Eq. (8):

$$J = \frac{\Delta V}{A \Delta t} \tag{8}$$

where $\Delta V$ (L) is the permeate volume during the running time $\Delta t$ (h), and $A$ was the effective separation area (m$^2$). The salt rejection ratio ($R$) was calculated with Eq. (9):

$$R = 1 - \frac{C_p}{C_f} \tag{9}$$

where $C_p$ and $C_f$ were the concentrations of salt ions in the permeate and the feed, measured with the conductivity meter (METTLER TOLEDO, FE38, Switzerland).

The mono/divalent anion selectivity (Cl$^-$/SO$_4^{2-}$) was measured by filtrating a mixed solution of Na$_2$SO$_4$ (0.5 g L$^{-1}$) and NaCl (0.5 g L$^{-1}$), and the cation selectivity (Li$^+$/Mg$^{2+}$) was detected by using a feed of 2.0 g L$^{-1}$ MgCl$_2$ and 0.1 g L$^{-1}$ LiCl (mass ratio 20:1). The ion concentrations were quantified by ionic chromatography (IC, Shimadzu, Japan). The selectivity of Cl$^-$ and SO$_4^{2-}$ was defined as follow:

$$S = \frac{(C_{Cl^-})_p/(C_{SO_4^{2-}})_p}{(C_{Cl^-})_f/(C_{SO_4^{2-}})_f} \tag{10}$$

and the selectivity of Li$^+$ and Mg$^{2+}$ was calculated as:

$$S = \frac{(C_{Li^+})_p/(C_{Mg^{2+}})_p}{(C_{Li^+})_f/(C_{Mg^{2+}})_f} \tag{11}$$

The separation ratio between small organic molecules and salt ions was also evaluated with the model feed (0.1 g L$^{-1}$ VB12 and 0.5 g L$^{-1}$ Na$_2$SO$_4$ or 0.5 g L$^{-1}$ NaCl). The PA-IL95% membrane was applied in the nanofiltration test. The concentration of VB12 was measured by UV spectroscopy and the ion concentrations were detected by IC. The ion/VB12 selectivity was calculated as Eq. (12):

$$S = \frac{(C_{ion})_p/(C_{VB12})_p}{(C_{ion})_f/(C_{VB12})_f} \tag{12}$$

The long-term durability of PA-IL membranes was evaluated by continuous crossflow nanofiltration of a binary salt solution (2.0 g L$^{-1}$ MgCl$_2$ and 0.1 g L$^{-1}$ LiCl) under 6 bar. Permeate flux and ion selectivity were measured every 12 h.

## Data availability
All data are available in the main text and Supplementary Information. All other data are available from the corresponding author upon request.

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

## Acknowledgements

We appreciate the financial support from the Natural Science Foundation of Zhejiang Province (Grant no. LD22E030001), the National Natural Science Foundation of China (Grant no. 22135006), the National Key Research & Development Program of China (Grant no. 2021YFB3801503), and the Fundamental Research Funds for the Central Universities (Grant no. 226-2023-00057). C.Z. and H.-C.Y. acknowledge gratefully research startup package from Zhejiang University. The authors would like to thank Yue-Qi Hu and Ye-Cheng Shen for their support in the experimental work.

## Author contributions

Z.-K.X., C.L. and B.-B.G. conceived the project. Z.-K.X. and C.Z. supervised the research. B.-B.G., C.L. and C.-Y.Z. prepared the materials, conducted the characterization, and analyzed the data. B.-B.G., C.L. and J.-H.X. performed molecular simulation. B.-B.G., C.L., C.Z., H.-C.Y. and Z.-K.X. wrote the paper, and all authors engaged in discussions related to the manuscript.

## Competing interests

The authors declare no competing interests.
