## [Peer Review File · Nature Communications]

Double charge flips of polyamide membrane by ionic liquid-decoupled bulk and interfacial diffusion for on-demand nanofiltrationReviewers' Comments:

Reviewer #1:

Remarks to the Author:

The work by Guo et al. employed ionic liquids (ILs) to decouple synchronous change of bulk/interfacial diffusion and therefore regulates the charge property of polyamide membranes. This strategy is intriguing; the experiment and characterizations are extensive. However, the data provided in this paper is not rigorous. In addition, introducing ILs into water as co-solvent has been reported by the same group (Angew. Chem. Int. Ed. 2021, 60, 2–9), where the mechanism of IP reaction at the Alkane/ILs interface and the properties of resulting PA membranes has been discussed in details. Additional comments are as follows:

1. In Line 146, "The theoretical bulk diffusion coefficient (D_b) of PIP is $3.58 \times 10^{-6} \text{ cm}^2 \text{ s}^{-1}$ and $2.11 \times 10^{-7} \text{ cm}^2 \text{ s}^{-1}$ in the 40 v/v% IL and 100 v/v% IL, respectively, both of which are much slower than that in pure water with a D_b of $9.30 \times 10^{-6} \text{ cm}^2 \text{ s}^{-1}$." Is the degree of this decline significant? Tan et.al reduced D_b by an order of magnitude using PVA (Science 2018, 360, 518–521).
2. It is expected that ILs enhance the interfacial diffusion of PIP like a surfactant. However, the interfacial TMC diffusion has also been increased as discussed (Angew. Chem. Int. Ed. 2021, 60, 2–9). Both of them influence the PIP supply in the reaction zone and impact the surface chargeability of the PA layer. Therefore, one can't only consider the effect of ILs on PIP alone.
3. In Line 209, it was stated that 'The PA-IL nanofilms are gradually flat and smooth with increased IL content'. Measurements of membrane roughness with AFM are needed to provide more evidence.
4. In Line 269, "The $\text{Mg}^{2+}/\text{Li}^{+}$ separation factors of PA-IL nanofiltration membranes follow the order PA-IL20% (9.0) < PA-IL40% (11.5) < PA-IL15% (18.6) < PA-IL10% (68.4)." Why the SLi, Mg of PA-IL40% possessing more positive charges is lower than that of PA-IL10% and PA-IL15% (Fig. 2b)? If the pore size plays a significant role in the separation mechanism, one should not just focus on charge regulation, or perhaps the variation in pore size is the key.
6. Two figures may be prepared to compare the SCI, SO_4 and SLi, Mg of the PA-IL membrane with state-of-the-art NF membranes to highlight the remarkable selectivity of the PA-IL membrane developed in this work.
7. One major concern is the reliability of SCI, SO_4 (407). As described in the manuscript, the mono/divalent anion selectivity ($\text{SO}_4^{2-}/\text{Cl}^-$) was measured by filtrating a mixed solution of Na_2SO_4 (0.5 g L⁻¹) and NaCl (0.5 g L⁻¹). The rejection of PA-IL5.0% to Na_2SO_4 is only 98.1% (Supplementary Table 7) in the single-salt separation and it is unusual that the mixed-salt selectivity can arrive as high as 407 in the mix-salt separation. The $\text{R}_{\text{Na}_2\text{SO}_4}$ and SCI, SO_4 of two literature reports which used same feed solution are listed in the table below. Their $\text{R}_{\text{Na}_2\text{SO}_4}$ is 99.6%, much higher than that of PA-IL5.0%, while SCI, SO_4 is less than 400.

Membranes $\text{R}_{\text{Na}_2\text{SO}_4}$ SCI, SO_4 Reference

SCOF/PA 99.6% 312.6 Chemical Engineering Journal 427 (2022) 132009

ULPA-2 99.6% 205.8 J. Mater. Chem. A, 2020,8, 23930-23938

Reviewer #2:

Remarks to the Author:

This work reported a facial ionic liquid-decoupled bulk/interfacial diffusion strategy to elaborate the double charge flips of polyamide membranes, allowing for on-demand transformation from inherently negative to highly positive and near-neutral charge. It is interesting and useful. Those charge-tunable polyamide membranes can be customized for separation performance in sulfate recovery, lithium extraction and pharmaceutical purification.

1. Normally, the oxygen nitrogen ratios corresponding to fully crosslinked and fully linear polyamides are 1:1 and 2:1, respectively. Generally, PIP/TMC crosslinks around 1.3:1. From S-table 1, the O/N ratio is 1.25, the membrane should be very dense or high crosslinked. Based on current in-depth analysis data, the O/N ratio is slightly greater than 1 in the absence of ion etching, slightly lower than

the conventional O/N ratio, and the positive charge should not be high; As the etching time prolongs, the O/N decreases significantly, reaching a minimum of 0.77. It should indeed show positive charge, but the crosslinking degree of the membrane should be very low, failing to form a well crosslinked structure. So how does the author explain it?

2. And membranes with different charges exhibit different performance in different fields, which is a conventional membrane performance and not very meaningful. The author should study the universality of this method, such as other additives in to PIP solution.

3. The author should focus on studying the first charge reversal process from negative to positive, especially to the first unlabeled "neutrally charged", and the research in the latter interval is obviously does not have much meaning, whether in theory or practical application. That is to say, replacing deionized water with IL at a cost of tens of millions of times higher, to prepare the reported nanofiltration membranes with good or ordinary performance for dye/salt separation.

4. According to the XPS analysis (Supplementary Table 1), there are certain IL residues on the membrane surface, which contain lots charged groups. Is the change in separation performance of these membrane related to these residual IL?

5. As shown in Supplementary Table 7, the PA-IL0% membrane, which is a nanofiltration membrane prepared by conventional interfacial polymerization (concluding vacuum assisted), has a MWCO of only 212 Da and exhibits a significant negative charge on its surface (-17.5 mV). However, although it has a high Na₂SO₄ rejection of 98.5%, it is still significantly lower than the literature results under similar conditions (generally higher than 99%, even compared to the results of author's research group). Although there is only a difference of about 1% in Na₂SO₄ rejection, this usually results in several times the pure water flux change. So, are there any issues with the author's membrane preparation process? If the author did it intentionally, please provide an explanation.

6. The author compared the membrane performance with the results of relevant literature, the latest relevant literature results should be added in Supplementary Figure 18.

Reviewer #3:

Remarks to the Author:

This work reports positive polyamide membranes by introducing ionic liquid as the co-solvent into the aqueous phase during interfacial polymerization. By altering the content of ionic liquid in the solution, the membrane surface charge can be well tuned from negative to positive and then back to nearly neutral. This leads to interesting performance in separating mono/di valent ions, especially enabling high selectivity of Mg²⁺/Li⁺. The authors should shift the focus of the paper to these intriguing results, as in the current format, the rationale of creating positively charged surfaces is unclear. For the paper to be published in Nature Communications, the manuscript should be revised to address the target separation challenge and the specific comments below.

1. Fig. 2c shows higher R₂NH content for PA-IL-40%, indicating a lower crosslinking degree, which results in faster permeance but higher MWCO, i.e., looser pores (Supplementary Table 6). On the other hand, the rejection of MgCl₂ increased for PA-IL-40% membranes as the surface charge became more positive (Supplementary Table 7). The authors should discuss more in details of the completing effect between pore size and surface charge on the solute rejections.

2. Interfacial polymerization is a self-limiting process, i.e., the growth of polyamide layer can prohibit or slow down the diffusion rate of amine. How does the reaction rate affect the diffusion of PIP with increasing IL content and thereby affecting the surface charge and membrane thickness? In this context, using 80 g.L⁻¹ of PIP (Line 190 Page 7) as a control experiment is inappropriate as the reaction rate would be so fast to form a dense polyamide layer that could prohibit/slow down further diffusion of PIP, resulting in less positively charged surface (Supplementary Figure 12).

3. How could the authors eliminate the interference of surface charge from ionic liquid as significant amount of fluorine was detected in the polyamide layer (Supplementary Table 1)?

4. The estimation of membrane pore size from PEG rejections is inappropriate since the conformation of PEG can be varied in different solvents or in the bulk versus at the membrane surface. Also, their molecular size has a broad distribution, for which using the mean solute size to estimate the

membrane pore size could be inaccurate and misleading.

5. Why did the PA-IL-10% membranes perform the best selectivity for Mg²⁺/Li⁺ separation (Fig. 5b) but the lowest permeance (Supplementary Table 8)? If the surface charge is the key for cation separations, membrane surfaces with higher positive charge at elevated ionic liquid content (e.g., PA-IL-40% membranes in Fig. 2b and Supplementary Table 6) should exhibit higher selectivity. Please justify.

6. Please add the data of LiCl rejection to Fig. 5a and Supplementary Table 7.

Reviewers' Comments:

Reviewer #1:

Remarks to the Author:

[Note from the Editor: Reviewer #1 was also asked to assess the response given to reviewer #2]:

The authors have responded to most questions. However, two questions raised by reviewer 2 has not been well addressed.

Comment 2: The reviewer commented that the O/N ratio of the membrane is only 0.77 and he/thinks the crosslinking degree of the membrane should be very low, failing to form a well-crosslinked structure. The authors replied they had put forward a new approach in another paper (in submission) to calculate the crosslinking degree. But the equations are difficult to understand (I couldn't understand). The authors should provided detailed explanations and equations in this manuscript. It is not reasonable to ask readers to wait for another paper to published before they could understand this paper.

Comment 3: The reviewer said 'And membranes with different charges exhibit different performance in different fields, which is a conventional membrane performance and not very meaningful. The author should study the universality of this method, such as other additives in to PIP solution.' The reviewer asked for the universality of their method. The authors used another ionic liquid ([Emim][BF₄]), similar to [Bmim][BF₄] that they used in this paper. The reviewers should employ other solvents and summarize the characteristics needed to arrive the 'charge flip' phenomenon.

Reviewer #3:

Remarks to the Author:

The revision based on the additional experiments and simulation shows convincing evidence and results, which have addressed all the critical comments. Therefore, I would recommend for publication.

[Note from the Editor: Reviewer #3 was also asked to assess the response given to reviewer #2]:

Whilst the authors answered most of the comments, the key question still remains as whether the membrane performance in this manuscript were significantly better/or ever better than the existing membranes reported in literature, in the compromise of using pricy ionic liquids.

If the different concentrations were selectively chosen for a fair comparison to the membranes made from IL system, what is the performance of the PIP membranes made from "common" conditions (i.e., membranes with >99% Na₂SO₄ rejections)? Is the selectivity of SO₄²⁻/Cl⁻¹ in IL-membranes still better than conventional PIP membranes?

In Supplementary Fig. 22, what is the parameter in x-axis at the bottom? If the figure is plotted against permeance, there is barely any advantage of the IL-membranes in this manuscript as compared to the reported membranes in literature, especially membranes made from novel monomers and substrate modification

Point-to-point response to the reviewers' comments

Reviewer #1:

Comment 1: The authors have responded to most questions. However, two questions raised by reviewer 2 has not been well addressed.

Response: We are pleased that the reviewer appreciates our responses. We will provide detailed answers to the two issues raised by the reviewer.

Comment 2: The reviewer commented that the O/N ratio of the membrane is only 0.77 and he/thinks the crosslinking degree of the membrane should be very low, failing to form a well-crosslinked structure. The authors replied they had put forward a new approach in another paper (in submission) to calculate the crosslinking degree. But the equations are difficult to understand (I couldn't understand). The authors should provide detailed explanations and equations in this manuscript. It is not reasonable to ask readers to wait for another paper to published before they could understand this paper.

Response: We thank the review for pointing out this crucial aspect. Generally, the PIP-TMC based polyamide membranes possess a complicated molecular structure comprising crosslinking structures (*X*), linear structures (*Y*) and two terminate structures with either amino (*T_{amino}*) or carboxyl groups (*T_{carboxyl}*), as depicted in Fig. R1.

Fig. R1 Schematic representation of all potential molecular structures in PIP-TMC polyamide membrane. Here, *X*, *Y*, *T_{amino}*, and *T_{carboxyl}* represent the crosslinking structure, linear structure, amino-terminated structure, and carboxyl-terminated structure, respectively.

Conventionally, the calculation of the degree of network crosslinking (DNC) includes only the crosslinking structures (X) and linear structures (Y), yet neglecting the terminal structures of T_{amino} or $T_{carboxyl}$:

$$DNC = \frac{X}{X + Y} \quad (R1)$$

Given that each X consists of three O atoms and three N atoms, and each Y has four O atoms and two N atoms, the O/N ratio can be expressed by the following equation:

$$\frac{O}{N} = \frac{3X + 4Y}{3X + 2Y} \quad (R2)$$

Consequently, DNC can be further expressed as:

$$DNC = \frac{4 - 2\frac{O}{N}}{1 + \frac{O}{N}} \quad (R3)$$

As a consequent, the membrane's crosslinking degree is evaluated using the O/N ratio on the basis of Equation R3. However, owing to the fact that each T_{amino} contributes to one N atom and each $T_{carboxyl}$ adds four O atoms, these N and O atoms have no contribution to cross-linking degree. Consequently, employing the total N and O atoms ratio from X-ray photoelectron spectroscopy (XPS) for DNC calculation lacks precision. This assertion is corroborated by several studies where DNC values of polyamide membranes are negative but still exhibit good separation performance (*Adv. Mater.* 2018, 30, 1705973; *ACS Appl. Mater. Interfaces* 2020, 12, 25304-25315; *Desalination* 2020, 491, 114345; and *J. Mater. Chem. A* 2020, 8, 3238-3245).

To address this issue, we propose that the concept of the amide group ratio ($r_{N-C=O}$) provides a more accurate depiction of the crosslinking degree or compactness in polyamide membranes. Here, $r_{N-C=O}$ is defined as the ratio of the number of amide groups (n_{amide}) in the polyamide membrane to the total number of functional groups (amide, amino, and carboxyl groups, shown in Fig R2):

$$r_{N-C=O} = \frac{n_{amide}}{n_{amide} + n_{amino} + n_{carboxyl}} \quad (R4)$$

From this equation, we found that a higher $r_{N-C=O}$ means more formation of amide groups, implying a higher crosslinking degree and increased membrane compactness.

Fig. R2 Schematic representation of all potential functional group structures in polyamide and their corresponding N and O atom counts.

Utilizing XPS data followed by the deconvolution of narrow N1s and O1s spectra, we can determine the $N_{(N-C=O)}$ and $O_{(N-C=O)}$ in amide groups, respectively. In Equation R4, n_{amide} is dictated by the $N_{(N-C=O)}$, while $n_{amide}+n_{amino}$ is represented by the total N content (N). $n_{carboxyl}$ is calculated as the total O content (O) minus the $O_{(N-C=O)}$:

$$n_{carboxyl} = \frac{O - O_{(N-C=O)}}{2} \quad (R5)$$

Thus, $r_{N-C=O}$ can be calculated from the following equation:

$$r_{N-C=O} = \frac{N_{(N-C=O)}}{N + \frac{O - O_{(N-C=O)}}{2}} = \frac{2N_{(N-C=O)}}{2N + O - O_{(N-C=O)}} \quad (R6)$$

Employing this method, we confirmed that the proportion of surface and internal amide bonds in the PA-IL40% membrane is approximately 80.3% and 70.9%, respectively (Supplementary Table 3). These values align with those in the PA-IL0% membrane (about 76%) known for its high crosslinking density. These results demonstrate that the PA-IL40% membrane also possesses a robustly crosslinked structure.

Supplementary Table 3 Elemental profile and the deconvoluted amount of amide groups in the inner PA-IL40% membrane.

Etch time (s)	N (%)	O (%)	N/O ratio	$O_{(N-C=O)}$ (%)	$r_{N-C=O}$ (%)
0	12.65	13.42	0.94	11.10	80.3
20	14.18	12.65	1.12	11.02	77.2
40	14.58	11.67	1.25	10.21	69.2
60	14.30	11.44	1.25	10.05	68.1
80	14.17	11.49	1.23	10.13	71.6
100	14.09	11.33	1.24	10.15	70.9

Following the reviewer's suggestion, we have detailed analytical method of the amide group ratio in the revised Supplementary Information.

Comment 3: The reviewer said 'And membranes with different charges exhibit different performance in different fields, which is a conventional membrane performance and not very meaningful. The author should study the universality of this method, such as other additives in to PIP solution.' The reviewer asked for the universality of their method. The authors used another ionic liquid ([Emim][BF₄]), similar to [Bmim][BF₄] that they used in this paper. The reviewers should employ other solvents and summarize the characteristics needed to arrive the 'charge flip' phenomenon.

Response: We agree with the reviewer that the universality of 'charge flip' phenomenon of polyamide membranes is much meaningful compared to the separation performance. Despite this, our IL-mediated charge flip approach demonstrates the capability of both positively and negatively charged polyamide membranes to exhibit remarkable monovalent/divalent ion selectivity over the conventional polyamide membranes. For instance, our PA-IL5% membrane delivers an outstanding Cl⁻/SO₄²⁻ selectivity exceeding 470, and our PA-IL10% achieves a Li⁺/Mg²⁺ selectivity greater than 68, surpassing many reported in existing studies (Supplementary Fig. 22).

Supplementary Fig. 22 Comparison of the $\text{Li}^+/\text{Mg}^{2+}$ selectivity and water permeability of PA-IL membranes with that state-of-art positively charged nanofiltration membranes. This includes membranes synthesized via one-step IP using PEI, novel monomers, or co-monomers, as well as membranes fabricated through multi-step IP featuring interlayer modification or post-grafting.

Next, we will answer the comment regarding the universality of the charge flip mechanism. To validate the universality of the charge flip mechanism, we incorporated sodium dodecyl sulfate (SDS) and glycerol as aqueous additives in the interfacial polymerization process, in which the former is capable of facilitating interfacial diffusion via the reduced interfacial tension and the latter can harness its intrinsic viscosity to suppress bulk diffusion. Consequently, we hypothesize that such an integrated strategy can effectively decoupling interfacial and bulk diffusion, similar to our IL-enabled strategy. Briefly, we employed an SDS concentration of 4.5 mM and added glycerol at a volume fraction of 17%, effectively replicating the interfacial tension and viscosity of a 40 v/v% [Bmim][BF₄]/water solution. The concentrations of PIP and TMC were maintained at 8 g L⁻¹ and 2.4 g L⁻¹, consistent with the conditions detailed in our manuscript. Under these specific conditions, the resulting polyamide membrane exhibits a zeta potential of approximately +27.8 mV at pH 6 (PA-Gly17%-SDS in Fig. R3), contrasting with previous studies where polyamide membranes produced using either

SDS or glycerol alone are negatively charged (*Nat. Commun.* 2020, 11, 2015 and *J. Membr. Sci.* 2021, 627, 119142). This finding underscores the importance of decoupled bulk/interfacial diffusion in achieving a charge flip of polyamide membranes.

Taken together, our current experimental and simulation results indicate that reducing interfacial tension and increasing bulk viscosity can achieve a decoupling of interfacial and bulk diffusion of amine monomer, leading to charge flip of polyamide membranes. This phenomenon demonstrates the universality for both alkyl imidazolium tetrafluoroborate ionic liquid/water solvent systems and the SDS/glycerol additive systems. From a broader perspective, we believe that this work offers an insightful direction for exploiting new charge flip of polyamide membranes by designing interfacial polymerization systems that are capable of reducing interfacial tension and increasing bulk viscosity simultaneously.

Following the reviewer's suggestion, we have included these results and discussions in the revised manuscript.

Fig. R3 Comparison of the zeta potential at pH 6 between PA-IL0%, PA-IL40%-[Emim][BF₄], PA-IL40%-[Bmim][BF₄], and PA-Gly17%-SDS.

Page 8: "More intriguingly, our decoupled bulk/interfacial diffusion mechanism to create positive charge of polyamide membrane is generic and can be also extended to other functional molecules. First, we explore this mechanism using 1-ethyl-3-methylimidazolium tetrafluoroborate ([Emim][BF₄]), a molecular analogue of

[Bmim][BF₄], as co-solvent to execute the similar interfacial polymerization. Remarkably, at a concentration of 40 v/v%, [Emim][BF₄] also induces a charge flip on the polyamide membrane, although less pronounced than with [Bmim][BF₄] (Supplementary Fig. 14). This variation is attributable to the physicochemical properties of [Emim][BF₄], exhibiting slightly higher surface tension ($57.5 \pm 0.3 \text{ mN m}^{-1}$) and lower viscosity ($1.72 \pm 0.02 \text{ mPa}\cdot\text{s}$). To further extend universality, we substituted ionic liquids with a combination of sodium dodecyl sulfate (SDS) and glycerol as additives in the interfacial polymerization, in which the former is capable of facilitating interfacial diffusion via the reduced interfacial tension and the latter can harness its intrinsic viscosity to suppress bulk diffusion. This approach also successfully achieves a charge flip from negative to positive on the polyamide membrane (Supplementary Fig. 14). These results strongly validate the decoupled bulk/interfacial diffusion mechanism as a reliable method for modulating the surface charge of PIP-based polyamide membranes.”

Reviewer #3:

Comment 1: The revision based on the additional experiments and simulation shows convincing evidence and results, which have addressed all the critical comments. Therefore, I would recommend for publication.

Response: We are grateful to the reviewer for his/her constructive feedback, which significantly contributed to the enhancement and refinement of our work. We also appreciate the positive assessment of our efforts and revisions.

Comment 2: Whilst the authors answered most of the comments, the key question still remains as whether the membrane performance in this manuscript were significantly better/or ever better than the existing membranes reported in literature, in the compromise of using pricy ionic liquids. If the different concentrations were selectively chosen for a fair comparison to the membranes made from IL system, what is the performance of the PIP membranes made from “common” conditions (i.e., membranes with >99% Na₂SO₄ rejections)? Is the selectivity of Cl⁻/SO₄²⁻ in IL-membranes still better than conventional PIP membranes?

Response: We appreciate the reviewer’s insightful comment. We agree with that it is not a fair comparison to the membranes made from different methods using the similar monomer concentrations. In our study, we have optimized the concentrations of PIP and TMC to be 8 g L⁻¹ and 2.4 g L⁻¹, respectively, which indeed are distinct from the optimal conditions for traditional alkane-water interfacial polymerization. In response to the reviewer’s suggestion, we have synthesized polyamide membranes using three typical monomer concentrations in alkane-water interfacial polymerization systems and have compared their selectivity of Cl⁻/SO₄²⁻ with our PA-IL membranes. Briefly, TMC concentration in hexane was fixed at 1.5 g L⁻¹, while the concentrations of PIP aqueous solutions were changed from 1.0 g L⁻¹ to 1.5 g L⁻¹, and 2.0 g L⁻¹.

The experimental results show that polyamide membranes synthesized using the above-mentioned concentrations for conventional interfacial polymerization all exhibit Na₂SO₄ rejection rates exceeding 99%, with NaCl rejection rates below 30% (Table R2). These membranes demonstrated Cl⁻/SO₄²⁻ selectivity ranging from 260 to 350, which were lower compared to our IL systems (Table R3, Fig. R4).

Fig. R4 Comparative illustration of Cl⁻/SO₄²⁻ selectivity in polyamide membranes synthesized from conventional interfacial polymerization (denoted as CIP-PIP followed

by PIP concentration in g L^{-1}) and those obtained from IL-based interfacial polymerization.

Table R2 Nanofiltration performance of polyamide membranes synthesized with different PIP concentrations and 1.5 g L^{-1} TMC in n-hexane.

PIP concentration (g L^{-1})	1000 ppm NaCl		1000 ppm Na_2SO_4	
	Water flux ($\text{L m}^2 \text{ h}^{-1} \text{ bar}^{-1}$)	Salt rejection (%)	Water flux ($\text{L m}^2 \text{ h}^{-1} \text{ bar}^{-1}$)	Salt rejection (%)
1.0	16.9 ± 2.3	18.2 ± 2.5	12.7 ± 1.4	99.0 ± 0.2
1.5	13.0 ± 1.6	21.7 ± 0.8	9.9 ± 1.3	99.2 ± 0.2
2.0	12.7 ± 2.1	28.3 ± 2.0	8.0 ± 0.9	99.2 ± 0.3

Table R3 Anion selectivity of polyamide membranes synthesized with different PIP concentrations and 1.5 g L^{-1} TMC in n-hexane. The mixed feed contains 500 ppm NaCl and 500 ppm Na_2SO_4 , with test concentrations of Cl^- and SO_4^{2-} at 314.2 and 353.0 mg L^{-1} , respectively.

PIP concentration (g L^{-1})	Cl^- (mg L^{-1})	SO_4^{2-} (mg L^{-1})	R (Cl^-) (%)	R (SO_4^{2-}) (%)	S ($\text{Cl}^-/\text{SO}_4^{2-}$)
1.5	271.3	1.06	13.7	99.70	287.7
2.0	274.5	0.88	12.6	99.75	349.6

Comment 3: In Supplementary Fig. 22, what is the parameter in x-axis at the bottom? If the figure is plotted against permeance, there is barely any advantage of the IL-membranes in this manuscript as compared to the reported membranes in literature, especially membranes made from novel monomers and substrate modification.

Response: We thank the reviewer for pointing out this issue. The initial Supplementary Fig. 22 was designed to illustrate only the $\text{Li}^+/\text{Mg}^{2+}$ selectivity of polyamide membranes

synthesized under various methods in the y-axis, yet did not encompass water permeability data in the x-axis. Consequently, the x-axis in the initial manuscript has no physical meaning. In response to the reviewer's suggestion, we have revised Supplementary Fig. 22 to represent both water permeability and ion selectivity of our PA-IL membranes and the previously reported works. We have also supplemented the relevant data in Supplementary Table 10. To our delight, our PA-IL membranes deliver a good water permeability of $1.45 \times 10^{-6} \text{ L m}^{-1} \text{ h}^{-1} \text{ bar}^{-1}$, along with an impressive $\text{Li}^+/\text{Mg}^{2+}$ selectivity of up to 68, surpassing the most of state-of-art positively charged nanofiltration membranes.

Supplementary Fig. 22 Comparison of the $\text{Li}^+/\text{Mg}^{2+}$ selectivity and water permeability of PA-IL membranes with that state-of-art positively charged nanofiltration membranes. This includes membranes synthesized via one-step IP using PEI, novel monomers, or co-monomers, as well as membranes fabricated through multi-step IP featuring interlayer modification or post-grafting.

Reviewers' Comments:

Reviewer #1:

Remarks to the Author:

The authors have addressed Comment 2 raised by reviewer 2; however, response to Comment 1 is still not convincing; in fact, it is scientifically incorrect.

The authors may misunderstand the traditional method of correlating XPS results with crosslinking degree. Usually, the method tries to push to two limits: fully crosslinked, and fully linear structure. In the fully crosslinked state, due to the large molecular weight (which is validated by the insolubility in solvents), the contribution of terminal groups to the overall elementary distribution is negligible.

In Fig. R1, the authors solely draw a small unit consisting of 3 TMC and 4 PIP monomers. The unit cannot even be called a polymer. If it continues to grow as a polymer, the terminal amine group and 1-2 of the terminal carboxylic acid group will disappear. The authors cannot establish an equation for this small unit to represent polymers.

Reviewer #3:

Remarks to the Author:

The authors have addressed all the comments raised by me (#Ref 3) and #Ref 2. I would suggest for publication in Nature Communications.

Reviewer #1:

Comment 1: The authors have addressed Comment 2 raised by reviewer 2; however, response to Comment 1 is still not convincing; in fact, it is scientifically incorrect.

Response: We appreciate the reviewer's acknowledgment of our efforts in addressing Comment 2. Following the reviewer's suggestion, we have made a more scientifically robust and comprehensible clarification to address the concerns raised.

Comment 2: The authors may misunderstand the traditional method of correlating XPS results with crosslink degree. Usually, the method tries to push to two limits: fully crosslinked, and fully linear structure. In the fully crosslinked state, due to the large molecular weight (which is validated by the insolubility in solvents), the contribution of terminal groups to the overall elementary distribution is negligible.

Response: In the traditional method, the degree of crosslinking in polyamide networks is assessed by considering both fully crosslinked and fully linear structures, as illustrated in Equation R1:

$$\text{crosslinking degree} = \frac{X}{X+Y} = \frac{4-2\frac{O}{N}}{1+\frac{O}{N}} \quad (\text{R1})$$

where X and Y denote the crosslinked and linear structures within polyamide, respectively (*Desalination* 2011, 278, 387-396; *Science*, 2015, 348, 1347-1351). This traditional view hypothesizes that the “fully crosslinked state” exhibits a minimal impact of terminal groups on the elemental composition of polyamide membranes. However, in actual polyamide membranes, the “fully crosslinked state” is more theoretical concept than an achievable reality. Achieving a completely crosslinked structure in polyamide membranes is nearly impossible due to inherent challenges, such as the hydrolysis of acyl chloride monomers and the steric hindrances. This assertion is supported by numerous studies reporting O/N ratios greater than 1.3, suggesting a crosslinking degree under 60% (Table R1). Notably, some studies report O/N ratios exceeding 2, which, in theory, would indicate a negative degree of crosslinking. Despite this, these polyamide membranes exhibit superior performance, with water flux rates

exceeding $30 \text{ L m}^{-2} \text{ h}^{-1} \text{ bar}^{-1}$ and Na_2SO_4 rejection rate above 99% (*J. Mater. Chem. A* 2020, 8, 3238-3245). These findings manifest that actual polyamide membranes significantly deviate from being fully crosslinked, challenging the traditional method's effectiveness in correlating the degree of crosslinking with membrane compactness and separation efficiency.

Table R1. Some reported O/N ratios, degree of crosslinking, and separation performance of polyamide membranes.

C (%)	O (%)	N (%)	O/N	DNC (%)	Na_2SO_4 rejection (%)	Water permeance ($\text{L m}^{-2} \text{ h}^{-1} \text{ bar}^{-1}$)	Ref.
70.97	18.15	10.88	1.67	24.9	98.0	5.3	Sep. Purif. Technol. 2021, 275, 119227
76.42	15.03	8.55	1.76	17.6	79.7	9.3	
69.21	19.61	11.18	1.75	17.9	90.0	7.0	
73.81	16.38	9.81	1.67	24.7	93.0	4.8	
79.31	13.37	7.16	1.87	9.3	84.0	2.5	
72.86	16.09	11.05	1.46	44.3	98.1	46.6	J. Membr. Sci. 2021, 634, 119450
71.19	17.62	11.2	1.57	33.2	92.0	55.0	
69.65	18.19	12.17	1.49	40.5	97.2	7.5	Desalination 2021, 512, 115118
70.21	16.73	13.06	1.28	63.0	99.2	13.4	
69.06	19.71	11.23	1.76	17.8	97.8	9.2	
69.05	18.99	11.95	1.59	31.7	98.4	15.4	
70.04	21.07	8.89	2.37	-22.0	94.0	50.0	J. Mater. Chem. A 2020, 8, 3238-3245
69.06	21.92	9.01	2.43	-25.2	91.0	55.0	
72.45	17.45	10.1	1.73	20.0	96.0	6.3	J. Membr. Sci. 2018, 550, 36-44
74.80	14.51	10.69	1.36	54.5	93.0	1.7	
66.82	17.67	10.58	1.67	24.7	99.5	27.0	ACS Appl. Mater. Interfaces 2020, 12, 25304-25315
56.37	20.52	8.91	2.30	-18.3	87.5	36.5	
75.26	21.16	3.59	5.89	-113.0	98.0	7.5	J. Membr. Sci. 2016, 515, 238-244
75.08	19.17	5.75	3.33	-61.6	97.6	17.5	
75.94	17.69	6.37	2.78	-41.1	94.2	12.7	
76.46	17.04	6.5	2.62	-34.3	95.8	10.5	
76.63	17.65	6.72	2.63	-34.6	94.8	8.7	

Moreover, considering that current characterization methods may not definitively ascertain whether carboxylic groups in polyamide network are integrated within linear structures or positioned terminally, we have confirmed the significant presence of

carboxylic terminal groups through molecular dynamics simulations. For instance, we constructed a model of a PIP-TMC polyamide membrane with a reaction degree of 95% using the modeling approach from our previous work (*Angew. Chem. Int. Ed.* 2021, 60, 14636). According to traditional calculations, the degree of crosslinking of this model stands at 93.6%, yet terminal groups account for 8.5% of the total functional groups and 83.6% of all unreacted groups. This underscores that even highly crosslinked polyamide networks may harbor a substantial number of terminal groups.

More importantly, given that amines are difunctional monomers, unreacted amine groups can only exist as terminal groups, unable to contribute to the formation of linear structures. This is fully validated by our IL-enabled polyamide membranes exhibiting pronounced positive charges derived from these terminal amine groups.

Taken together, these results all strongly demonstrate that the contribution of terminal groups to the crosslinking degree of polyamide should not be negligible.

Fig. R1 Full-atom molecular model of a PIP-TMC membrane under the reaction degree of 95%. The amine and carboxylic terminal groups are represented with a bold ball-and-stick model, where carbon, oxygen, nitrogen, and hydrogen are depicted in cyan, red, blue, and white, respectively.

Comment 3: In Fig. R1, the authors solely draw a small unit consisting of 3 TMC and 4 PIP monomers. The unit cannot even be called a polymer. If it continues to grow as a polymer, the terminal amine group and 1-2 of the terminal carboxylic acid group will disappear. The authors cannot establish an equation for this small unit to represent polymers.

Response: We are grateful to the reviewer for pointing out this issue that could potentially lead to misunderstanding. We have updated Fig. R2 to more accurately depict all conceivable chemical structures within the polyamide, rather than using a mere small molecule in our previous version.

Fig. R2 Schematic representation of all potential chemical structures in PIP-TMC polyamide membrane. Here, *X*, *Y*, *T_{amino}*, and *T_{carboxyl}* represent the crosslinked structure, linear structure, amino-terminated structure, and carboxyl-terminated structure, respectively.

To answer the reviewer's concern that "If it continues to grow as a polymer, the terminal amine group and 1-2 of the terminal carboxylic acid group will disappear", we have checked the XPS data of IL-enabled polyamide membranes. Obviously, polyamide typically retains a certain number of carboxyl and amino groups (Supplementary Fig. 4). This result indicates the presence of numerous terminal amine and carboxyl group. The presence of terminal amine and carboxyl group can be further understood from the possible reaction processes during interfacial polymerization. On the one hand, once acyl chloride is hydrolyzed to a carboxyl group, it can no longer react with amines under interfacial polymerization conditions. On the other hand,

amines may also become unreactive, either due to their quaternization or because acyl chloride cannot reach the reaction zone.

Supplementary Fig. 4 Deconvolution of N1s and O1s spectra along the polyamide profile of the PA-IL40% membrane.

To provide an accurate representation of the compactness of polyamide membranes, especially considering terminal groups—a critical aspect we have previously emphasized—we introduce the concept of the amide group ratio ($r_{N-C=O}$). This ratio is defined as the number of amide groups within the polyamide membrane relative to the total number of amide (n_{amide}), amino (n_{amino}), and carboxyl ($n_{carboxyl}$) groups (Fig. R3), providing a precise metric for assessing crosslinking density.

$$r_{N-C=O} = \frac{n_{amide}}{n_{amide} + n_{amino} + n_{carboxyl}} \quad (R2)$$

The quantification for n_{amide} , n_{amino} , and $n_{carboxyl}$ is derived from XPS data, following the deconvolution of narrow N1s and O1s spectra. Specifically, n_{amide} is determined by the $N_{(N-C=O)}$:

$$n_{amide} = N_{(N-C=O)} \quad (R3)$$

while the sum of n_{amide} and n_{amino} is reflected in the total nitrogen content (N):

$$n_{amide} + n_{amino} = N \quad (R4)$$

$n_{carboxyl}$ is calculated from the total oxygen content (O) minus the oxygen associated with N-C=O bonds ($O_{(N-C=O)}$):

$$n_{carboxyl} = \frac{O - O_{(N-C=O)}}{2} \quad (R5)$$

Thus, $r_{N-C=O}$ can be calculated from the following equation:

$$r_{N-C=O} = \frac{N_{(N-C=O)}}{N + \frac{O - O_{(N-C=O)}}{2}} = \frac{2N_{(N-C=O)}}{2N + O - O_{(N-C=O)}} \quad (R6)$$

Using this method, we confirmed that the surface and internal amide bond proportions in the PA-IL40% membrane are approximately 80.3% and 70.9%, respectively, as detailed in Supplementary Table 3. These percentages are comparable to those observed in the PA-IL0% membrane, known for its high crosslinking density, thus evidencing the robustly crosslinked structure of the PA-IL40% membrane.

Supplementary Fig. R3 Schematic representation of all potential functional groups in polyamide and their corresponding N and O atom counts.

Supplementary Table 3 Elemental profile and the deconvoluted amount of amide groups in the inner PA-IL40% membrane.

Etch time (s)	N (%)	O (%)	N/O ratio	$O_{(N-C=O)}$ (%)	$r_{N-C=O}$ (%)
0	12.65	13.42	0.94	11.10	80.3

20	14.18	12.65	1.12	11.02	77.2
40	14.58	11.67	1.25	10.21	69.2
60	14.30	11.44	1.25	10.05	68.1
80	14.17	11.49	1.23	10.13	71.6
100	14.09	11.33	1.24	10.15	70.9

In response to the reviewer's suggestion, we have also revised Supplementary Fig. A1 and Fig. A2 in the Supplementary Information to ensure a clearer understanding of our methods.

Reviewers' Comments:

Reviewer #1:

Remarks to the Author:

The reviewer has no more comment, except that in the traditional model, in the status between fully crosslinked and linear structure, the presence of free amine groups and carboxylic acid groups are included in the equation.

Reviewer #1:

Comment 1: The reviewer has no more comment, except that in the traditional model, in the status between fully crosslinked and linear structure, the presence of free amine groups and carboxylic acid groups are included in the equation.

Response: We thank the reviewer for their detailed review and recognition of our response. In the traditional method, the degree of network crosslinking (DNC) in polyamide networks includes only the crosslinking structures (X) and linear structures (Y). However, as shown in Supplementary Fig. A1, this calculation method indeed neglects the contribution of the terminal structures of T_{amino} or $T_{carboxyl}$ (*Desalination* 2011, 278, 387-396; *Science*, 2015, 348, 1347-1351).

$$\text{DNC} = \frac{X}{X + Y} \quad (\text{R1})$$

Given that each X consists of three O atoms and three N atoms, and each Y has four O atoms and two N atoms, the O/N ratio can be expressed by the following equation:

$$\frac{O}{N} = \frac{3X + 4Y}{3X + 2Y} \quad (\text{R2})$$

Consequently, DNC can be further expressed as:

$$\text{DNC} = \frac{4 - 2\frac{O}{N}}{1 + \frac{O}{N}} \quad (\text{R3})$$

As a consequent, the membrane's DNC is evaluated using the O/N ratio on the basis of Equation R3. However, owing to the fact that each T_{amino} contributes to one N atom and each $T_{carboxyl}$ adds four O atoms, these N and O atoms have no contribution to cross-linking degree. Consequently, employing the total N and O atoms ratio from X-ray photoelectron spectroscopy (XPS) for DNC calculation lacks precision. This assertion is corroborated by several studies where DNC values of polyamide membranes are negative but still exhibit good separation performance (*Desalination* 2021, 512, 115118; *ACS Appl. Mater. Interfaces* 2020, 12, 25304-25315; and *J. Mater. Chem. A* 2020, 8, 3238-3245).

Supplementary Fig. A1 Schematic representation of all potential chemical structures in PIP-TMC polyamide membrane. Here, *X*, *Y*, *T_{amino}*, and *T_{carboxyl}* represent the crosslinked structure, linear structure, amino-terminated structure, and carboxyl-terminated structure, respectively.